# An Outcome Assessment of a Single Institution’s Longitudinal Experience with Uveal Melanoma Patients with Liver Metastasis

**DOI:** 10.3390/cancers12010117

**Published:** 2020-01-01

**Authors:** Rino S. Seedor, David J. Eschelman, Carin F. Gonsalves, Robert D. Adamo, Marlana Orloff, Anjum Amjad, Erin Sharpe-Mills, Inna Chervoneva, Carol L. Shields, Jerry A. Shields, Michael J. Mastrangelo, Takami Sato

**Affiliations:** 1Department of Medical Oncology, Sidney Kimmel Cancer Center, Thomas Jefferson University, Philadelphia, PA 19107, USA; 2Department of Radiology, Thomas Jefferson University, Philadelphia, PA 19107, USA; 3Department of Pharmacology and Experimental Therapeutics, Thomas Jefferson University, Philadelphia, PA 19107, USA; 4Oncology Service, Wills Eye Hospital, Philadelphia, PA 19107, USA

**Keywords:** uveal melanoma, metastatic uveal melanoma, liver metastasis, survival, treatment strategy, liver-directed treatment, real-world data

## Abstract

There is no FDA-approved treatment for metastatic uveal melanoma (UM) and overall outcomes are generally poor for those who develop liver metastasis. We performed a retrospective single-institution chart review on consecutive series of UM patients with liver metastasis who were treated at Thomas Jefferson University Hospital between 1971–1993 (Cohort 1, *n* = 80), 1998–2007 (Cohort 2, *n* = 198), and 2008–2017 (Cohort 3, *n* = 452). In total, 70% of patients in Cohort 1 received only systemic therapies as their treatment modality for liver metastasis, while 98% of patients in Cohort 2 and Cohort 3 received liver-directed treatment either alone or with systemic therapy. Median Mets-to-Death OS was shortest in Cohort 1 (5.3 months, 95% CI: 4.2–7.0), longer in Cohort 2 (13.6 months, 95% CI: 12.2–16.6) and longest in Cohort 3 (17.8 months, 95% CI: 16.6–19.4). Median Eye Tx-to-Death OS was shortest in Cohort 1 (40.8 months, 95% CI: 37.1–56.9), and similar in Cohort 2 (62.6 months, 95% CI: 54.6–71.5) and Cohort 3 (59.4 months, 95% CI: 56.2–64.7). It is speculated that this could be due to the shift of treatment modalities from DTIC-based chemotherapy to liver-directed therapies. Combination of liver-directed and newly developed systemic treatments may further improve the survival of these patients.

## 1. Introduction

Although uveal melanoma (UM) is rare, comprising less than 5% of all melanomas, it is the most common primary intraocular malignancy in adults [1,2]. It is estimated that the incidence in the United States is approximately 5.2 per million population [3]. Despite successful treatments of the primary tumor, up to 50% of affected patients will develop systemic metastasis, with the liver involved in up to 90% of those patients [4,5]. Outcomes are poor for those who develop metastasis, with one-year survival reported to be 15% and median survival ranging from 4 to 15 months [5,6,7,8,9]. 

Our Ocular Oncology team at Wills Eye Hospital have previously published several large retrospective reviews evaluating clinical factors predictive of metastasis [10,11,12]. They found that patient age, ciliary body location, increasing tumor diameter, increasing tumor thickness, darkly pigmented tumor, and the presence of subretinal fluid, intraocular hemorrhage, or extraocular extension were correlated with increasing risk for metastasis [10]. The risk for metastasis and death also increased 2-fold with each increasing tumor size category in the American Joint Committee on Cancer 7th edition [11]. Recently, cytogenetic and molecular pathway profiles have become important as uveal melanoma is now generally classified into two basic risk categories. Patients are considered high risk if they have monosomy 3 (and 8q amplification) or gene expression profile (GEP) class 2. They are considered low risk if they have disomy of chromosomes 3, 6, and 8, as well as GEP class 1 [13].

Currently, there is no Food and Drug Administration (FDA) approved treatment for metastatic UM. Chemotherapeutic drugs, commonly used for the treatment of metastatic cutaneous melanoma, rarely induces responses in patients with UM, especially in the setting of metastatic disease to the liver. In particular, dacarbazine (DTIC)-based chemotherapies are reportedly ineffective in the treatment of metastatic UM [14,15]. Systemic treatment with a variety of agents have been tried including anti-angiogenics, kinase inhibitors, and more recently immunotherapies [6,16,17,18]. Although some small clinical studies have reported encouraging response rates with various survival outcomes, none have proceeded to a practice changing phase III trial. A recent meta-analysis of 29 phase II trials for metastatic UM conducted between 2000 and 2016 found that both progression-free survival (PFS) and overall survival (OS) remain poor (PUMMA analysis) [19,20]. The median PFS was 3.3 months (6-month PFS 27%), and the median OS was 10.2 months (1-year OS 43%) [19,20].

Despite dramatic improvement in outcomes for patients with advanced cutaneous melanoma with recent innovations in immunotherapy, a similar clinical benefit has not been seen in metastatic UM [8,16]. Objective response rates appear low in retrospective studies, ranging from 5% with ipilimumab, 3.6% with anti-programmed cell death 1 (PD-1) antibodies, and 10–21% with combination ipilimumab and nivolumab [8,16,21,22,23,24]. A recent literature review by Schank et al. summarizes these findings [16]. Two single-arm phase II studies of ipilimumab showed a median OS of 9.8 months (Spanish Melanoma Group study) and 6.8 months (DeCOG study) [25,26]. A multicenter retrospective series of 56 patients with metastatic UM who were treated with a PD-1 or PDL-1 inhibitor of pembrolizumab, nivolumab, or atezolizumab (62% had received prior ipilimumab) showed an overall response rate of only 3.6% with a median OS of 7.6 months [22]. As for combination immunotherapy, we are awaiting the final analysis of phase II studies of nivolumab with ipilimumab for untreated (NCT02626962) and treated (NCT0158194) metastatic UM patients. The interim analysis of the phase II trial presented at the European Society for Medical Oncology 2018 Congress showed an overall response rate of 12%, PFS of 3.3 months, and median OS of 12.7 months in previously untreated UM patients [27]. There was a retrospective multi-center study of 64 patients who had received ipilimumab with a PD-1 inhibitor which showed best overall response rate of 15.6% with median PFS and OS of 3.0 months and 16.1 months, respectively [24]. Grade 3 or 4 severe, treatment-related adverse events were seen in 39.1% of the patients [24]. Another retrospective nationwide population-based study in Denmark evaluated survival before (pre-ICI, *n* = 32) and after (post-ICI, *n* = 94) the approval of first-line treatment with immune checkpoint inhibitors (20% with ipilimumab with nivolumab). Median PFS improved from 2.5 months to 3.5 months [23]. The estimated one-year OS rate increased from 25.0% to 41.9% and the median OS improved from 7.8 months to 10.0 months, respectively [23]. 

The results of targeted therapies have overall been disappointing. A recent review by Croce et al. summarizes the most recent studies of targeted therapies in UM [28]. There was a systematic review of three open-label phase II, two open-label phase I, and one placebo-controlled phase III trial of different MEK inhibitors showing median PFS ranging from 3.1 weeks to 16 weeks [29]. The phase III clinical trial of selumetinib in UM failed to show a clinical benefit in combination with DTIC in 14 metastatic UM patients with a median OS of 19 months, although it was small patient population and confounders might have resulted in a patient selection bias [30]. 

Tebentafusp (IMCgp100) is a bispecific molecule comprised of a targeting end that constitutes a soluble T cell receptor targeting glycoprotein 100 (gp100), melanoma/melanocyte-related antigen, and an effector end targeting CD3, that is also being evaluated in metastatic UM patients [31]. Damato et al.’s recent review of Tebentafusp summarizes the update on the drug [31]. The first in-human study in both UM (14 patients) and cutaneous melanoma (33 patients) showed promise in UM with two partial responses and disease control in 8 patients [32]. In a subsequent phase I trial of 19 metastatic UM patients treated on the dose escalation portion of the study (heavily pre-treated with various agents including immunotherapy), the one-year OS was as high as 74% and the median OS has not yet been met [33]. Partial response was achieved in 3 (18%) patients and disease control was achieved in 11 (65%) patients [33]. 

At our institution, recognition of the poor prognosis associated with liver metastasis has led to the use of various liver-directed treatments including: immunoembolization (IE) with granulocyte-macrophage colony-stimulating factor (GM-CSF) +/− interleukin-2 (IL-2) [34,35,36], transarterial chemoembolization (TACE) with carmustine (BCNU) [37,38], radioembolization (RE) with Yttrium 90 resin microspheres [39], and drug-eluting beads with doxorubicin (DEBDOX) [40]. Prior to the introduction of these therapies, the majority of our patients were receiving DTIC-based systemic chemotherapies with dismal results consistent with published data from other institutions [14]. The purpose of this study is to analyze OS of UM patients with liver metastases at a single institution before and after the shift of treatment modalities from systemic chemotherapy to liver-directed approaches. This is one of the largest real-world data sets on metastatic UM patients who have received newly developed liver-directed treatments.

## 2. Results

Data were collected from 108 patients in the 1971–1993 database (Cohort 1), 201 patients in the 1998-2007 database (Cohort 2), and 456 patients in the 2008–2017 database (Cohort 3). There were 19 patients, 0 patients, and 3 patients in the 1971–1993, 1998–2007, and 2008–2017 database, respectively, that were excluded from the study due to loss of follow-up and lack of survival information. In the 1971–1993 database, there were 9 patients who did not receive treatment prior to death, while 3 patients and 1 patient in the 1998–2007 and 2008–2017 database, respectively, did not receive any treatment. After applying both inclusion and exclusion criteria as stated in the Method section, 80 patients in Cohort 1, 198 patients in Cohort 2, and 452 patients in Cohort 3 were selected for further analysis. Median follow up from initial eye diagnosis was 40.8 months (range 6.4–241.5) for Cohort 1, 62.6 months (range 8.2–454.7) for Cohort 2, and 55.4 months (range 4.1–338.3) for Cohort 3. Median follow up from liver metastases was 5.3 months (range 0.2–35.0) for Cohort 1, 13.6 months (range 2.0–179.4) for Cohort 2, and 17.0 months (range 1.1–113.3) for Cohort 3. All of the patients in Cohort 1 eventually died of metastatic UM. Only one patient was alive and censored in Cohort 2, while 78 patients were alive and censored in Cohort 3.

Table 1 summarizes the baseline characteristics of each group. A comparison of cohorts revealed significant differences in age at eye diagnosis (medians 60, 53, and 57 years, respectively, *p* = 0.005) and time from eye diagnosis to liver metastasis (medians 35.9, 41.7, and 35.6 months, respectively, *p* = 0.005). There were no statistical differences in gender, T-stage, and primary tumor location. Meanwhile, the differences in treatment modalities (for eye tumor, following adjuvant treatment, and treatment for liver metastasis) were significant (*p* < 0.001 for all, Table 1). In particular, the majority of Cohort 1 (58%) underwent enucleation while the majority of Cohorts 2 and 3 (58% and 67%, respectively) underwent plaque radiotherapy. The differences in adjuvant treatment are largely attributed to the 11% of Cohort 3 patients treated with sunitinib, while this treatment was not used in Cohorts 1 and 2.

When metastasis to the liver developed, in Cohort 1 there were 25 (31%) patients who had concurrent metastasis to other organs. In Cohort 2 and 3 there were 56 (28%) and 102 (23%) patients, respectively, with concurrent metastasis to other organs. Most commonly concurrent metastasis occurred in the lung, lymph nodes, and bone.

Once metastasis to the liver developed, the majority of Cohort 1 patients (70%) received DTIC-based systemic treatment as their first and second (if given) treatment modalities (Table 1). Meanwhile, the majority of Cohort 2 and 3 patients (70% and 56%, respectively) received only liver-directed treatment as their first and second (if given) treatment modalities. “Liver-directed + Systemic” therapies, concurrently or in sequence as their first two treatment modalities were given to 10%, 28%, and 42% of patients in Cohort 1, 2, and 3, respectively. As a result, 98% of patients in Cohort 2 and 3 received liver-directed treatment either alone or with systemic therapy as their first and second (if given) treatment modalities, while only 30% of patients in Cohort 1 received liver-directed treatment. The type of systemic and liver-directed treatments given to each Cohort of patients can be found in Appendix A, and the breakdown by their first and second treatment are shown in Table 2. 

Of the 141 patients who had available chromosomal analysis in Cohorts 2 and 3, 82% had monosomy-3, 6% had 6q loss, 57% had 8q amplification. Of the 41 patients who had molecular analysis in Cohorts 2 and 3, 88% were classified as DecisionDx-UM Class 2.

The Kaplan–Meier curves for Mets-to-Death OS and Eye Tx-to-Death OS are shown in Figure 1. The overall differences between cohorts were significant for Mets-to-Death OS and Eye Tx-to-Death OS (*p* < 0.001 for both endpoints).

Table 3 shows yearly Mets-to-Death and Eye Tx-to-Death survival rates in each cohort. The median Mets-to-Death OS and Eye Tx-to-Death OS in each cohort with 95% confidence intervals (CI) are shown in Table 4. 

The Mets-to-Death OS was significantly different between each pair of cohorts (*p* < 0.001 for Cohort 1 vs. 2 and Cohort 1 vs. 3; *p* = 0.010 for Cohort 2 vs. 3). The median Mets-to-Death OS was shortest in Cohort 1 (5.3 months, 95% CI: 4.2–7.0), longer in Cohort 2 (13.6 months, 95% CI: 12.2–16.6) and longest in Cohort 3 (17.8 months, 95% CI: 16.6–19.4) (Table 4). The one-year and two-year Mets-to-Death survival rates were dramatically different between Cohort 1 (one-year OS: 23%, 95% CI: 15–34%; two-year OS: 8%, 95% CI: 3–16%) and Cohort 2 (one-year OS: 59%, 95% CI: 52–66%; two-year OS: 28%, 95% CI: 22–35%), and between Cohort 1 and Cohort 3 (one-year OS: 67%, 95% CI: 63–72%; two-year OS: 35%, 95% CI: 31–40%) (Table 3). All patients in Cohort 1 died in 3 years. Cohort 3 had slightly higher Mets-to-Death survival rates in years 1–5 as compared to Cohort 2 (Table 3). This trend was significant according to the overall log-rank test (*p* = 0.010).

The Eye Tx-to-Death OS was significantly different between Cohort 1 and Cohort 2 (*p* < 0.001) and between Cohort 1 and Cohort 3 (*p* < 0.001), but not between Cohort 2 and Cohort 3 (*p* = 0.685). The median Eye Tx-to-Death OS was the shortest in Cohort 1 (40.8 months, 95% CI: 37.1–56.9), and similar in Cohorts 2 (62.6 months, 95% CI: 54.6–71.5) and Cohort 3 (59.4 months, 95% CI: 56.2–64.7) (Table 4). The Eye Tx-to-Death survival rates were consistently higher for later treated Cohorts 2 and 3 vs. Cohort 1, but the 95% confidence intervals overlap in follow-up years 1–7. However, long-term survival rates in follow-up years 8–10 were significantly lower for Cohort 1 (e.g., 8-year OS: 11%, 95% CI: 6–21%) as compared to Cohort 2 (e.g., 8-year OS: 31%, 95% CI: 25–38%) or Cohort 3 (e.g., 8-year OS: 27%, 95% CI: 23–32%) (Table 3).

Table 5 shows the results from the parsimonious Cox models of Mets-to-Death OS. In Cohort 1, Liver-directed + Systemic treatment (“Ever”, Treatment 1 + 2) was associated with longer Mets-to-Death OS as compared to Systemic only treatment (Hazard Ratio (HR) = 0.28, 95% CI: 0.12–0.65; *p* = 0.003) and as compared to Liver-directed only treatment (HR = 0.35, 95% CI: 0.14–0.86; *p* = 0.023). In other words, for patients who received liver-directed and systemic treatments in consecutive order or in combination, the hazard of death was 65% and 72% lower than for patients who received liver-directed therapy only or systemic therapy only, respectively. The difference between Systemic and Liver-directed only treatments was not significant. In addition to liver metastasis treatment, log-transformed Time from Initial Eye Treatment to Metastasis was a significant predictor of Mets-to-Death OS in Cohort 1. Two-fold increase in the Eye Treatment to Metastasis Time was associated with 17% decrease in hazard of death (HR = 0.83, 95% CI: 0.70–0.97; *p* = 0.022). Thus, longer time between eye treatment and metastasis implied longer OS after development of metastasis. For example, patients who developed metastasis 6 months after their eye treatment had a 17% lower hazard of dying as compared to patients who developed metastasis 3 months after eye treatment. In Cohort 2, Liver-directed + Systemic treatment was associated with 42% decrease in hazard of death (HR = 0.58, 95% CI: 0.42–0.80; *p* = 0.001) as compared to Liver-directed only treatment. There was also 35% lower hazard of death (HR = 0.65, 95% CI: 0.48–0.87; *p* = 0.003) for females as compared to males. In Cohort 3, the Mets-to-Death OS was also significantly different between Liver-directed + Systemic and Liver-directed only treated patients (*p* < 0.001), but the proportional hazard assumption was violated and survival differences were only observed in early follow-up years and diminished after 4 years (Appendix A).

Parsimonious Cox models of Eye Tx-to-Death OS were also performed (Appendix A). Only older age (60+ years old) was a significant predictor of shorter Eye Tx-to-Death OS (HR = 1.61, 95% CI: 1.01–2.56; *p* = 0.046) in Cohort 1. Similar results for older age were observed in Cohort 2 (HR = 1.52, 95% CI: 1.01–2.28; *p* = 0.044). Moreover, in Cohort 2, patients with ciliary body involvement had significantly shorter Eye Tx-to-Death OS (HR = 1.67, 95% CI: 1.10–2.54; *p* = 0.016), compared to that of patients with choroidal melanoma. Females had longer Eye Tx-to-Death OS (HR = 0.59, 95% CI: 0.41–0.87; *p* = 0.007) as compared to males. In Cohort 3, the Eye Tx-to-Death OS was significantly shorter for T-stages 3 and 4 as compared to T-stage 1. Older age was again associated with shorter Eye Tx-to-Death OS (HR = 1.93, 95% CI: 1.53–2.44; *p* < 0.001). The gender difference in Eye Tx-to-Death OS was also significant (*p* = 0.002), but the proportional hazard assumption was violated and survival differences were only observed in follow-up years 1–10 and diminished afterwards (Appendix A).

## 3. Discussion

In this retrospective study, we were able to demonstrate prolongation of survival between an earlier cohort and more recent cohorts of UM patients with liver metastasis. The median Mets-to-Death OS was shortest in Cohort 1 (5.3 months, 95% CI: 4.2–7.0), longer in Cohort 2 (13.6 months, 95% CI: 12.2–16.6) and longest in Cohort 3 (17.8 months, 95% CI: 16.6–19.4). Each database contained consecutive patients with liver metastasis seen at Thomas Jefferson University during that period and there was no selection bias of patients before being registered into the database. The three cohorts were similar in terms of demographic and eye tumor characteristics except for age at eye diagnosis. Nonetheless, we believe that our study is one of the largest, “real-world” data showing improvement of the outcomes of metastatic UM patients with liver metastasis over the last 47 years. 

Although direct comparison is not appropriate, our real-world data from a large cohort of UM patients with liver metastasis (*n* = 730) is comparable to the size of the PUMMA analysis (*n* = 912) [19,20], and should be considered as another landmark for future treatments of metastatic uveal melanoma. However, it should be recognized that the PUMMA analysis included all metastases (not limited to the liver) and their survival outcomes (10.2 months) were calculated from the date of first treatment for metastasis as opposed to the date of diagnosis of liver metastasis in our study. The PUMMA analysis was also based on prospective clinical trials at multiple centers.

Rantala et al. performed a meta-analysis with pooled data from 2494 patients from 78 articles on metastatic uveal melanoma (47% prospective, 52% retrospective) [41]. The median OS across all treatment modalities after metastasis was 1.07 years. Pooled median OS was longer than that of conventional chemotherapy (0.91 years) after isolated hepatic perfusion with a median OS of 1.34 years (HR = 0.92, 95% CI: 0.87–0.97; *p* = 0.004), 1.63 years after IE (HR = 0.97, 95% CI: 0.95–1.00; *p* = 0.008), and 1.43 years after surgery (HR = 0.94, 95% CI: 0.92–0.96, *p* < 0.001). They found that median OS after checkpoint inhibitor was only 0.59 years which was shorter than the 0.91 years with conventional chemotherapy (HR = 1.13, 95% CI: 1.06–1.20; *p* < 0.001), but they noted that the analysis is subject to identifiable confounding factors including the rarity of first-line treatments with immunotherapy [41].

Another single-institution report was recently published by Lane et al. in *JAMA Ophthalmology* in 2018 [9]. They compared outcomes of 661 patients with metastatic UM treated between 1982 and 2009 with the previous cohort of 145 patients treated between 1975 and 1987. Overall, they found outcomes were still quite poor with a median OS of 3.9 months following diagnosis of metastatic disease, with a better prognosis in those who received treatment (6.3 months vs. 1.7 months). In their earlier cohort of patients, the median OS was 5.2 months for patients who received treatment and 2 months for those who did not [42]. Of note, only 344 (53%) of the new cohort patients received treatment for metastasis and information regarding specific treatments were only available for 103 (30%) patients. Of those, 53 (50%) patients received chemotherapy. A much small number of patients (18 patients, 18%) underwent liver-directed therapies including intrahepatic arterial infusion or isolated hepatic perfusion (10 patients, 10%), and chemoembolization (8 patients, 8%). This is comparable to our Cohort 1 patients, most of whom were treated with DTIC-based systemic treatments alone (56 patients, 70%) with similar median OS after development of metastasis (5.3 months). In this regard, there was significant improvement in survival of our Cohort 2 and 3 patients (13.6 months and 17.8 months respectively), felt to be from the shift of treatment strategy away from DTIC-based systemic chemotherapy to liver-directed treatments.

Most recently, a retrospective nation-wide analysis of metastatic UM patients registered in the Dutch Melanoma Treatment Registry between July 2012 and March 2018 was reported by Jochems et al. [43] They did not report median overall survival, but for their 175 patients, survival rates at one-year in patients receiving systemic therapy or local therapy was 49% and 82.1%, respectively. 47.8% of all patients were alive one year after the diagnosis of metastatic UM, which is more than our Cohort 1 (23%) but less than our Cohort 2 (59%) and Cohort 3 (67%). Thirty-nine patients (22.3%) in their study received local therapy such as surgical resection, isolated hepatic perfusion with melphalan, radiotherapy, radiofrequency ablation or radio-embolization as their initial treatment [43], similarly to our Cohort 1 (30%).

Our results are consistent with an analysis performed by Moser et al., who found in their multivariate analysis of 101 metastatic UM patients diagnosed at the Mayo Clinic between 2000 and 2013 that treatment with local liver therapy significantly improved OS (risk ratio 0.23, *p* < 0.001) [44]. Local liver therapy (surgical resection, radiation, radiofrequency ablation, TACE, IE, RE, and intra-arterial chemotherapy) was associated with an impressive increased median OS of 26 months (*n* = 46) compared to 9.1 months (*n* = 55) in those who received no liver-directed therapy. Survival also increased with the number of local therapies that individual patients received. They do acknowledge that these patients had better markers of prognosis at the time of treatment. Ipilimumab, kinase inhibitors, and bevacizumab all had a positive effect on survival but without statistical significance.

There is potential criticism that the OS of Cohort 2 and 3 may appear improved because of lead-time bias. Early discovery of small metastasis by modern technology might result in longer OS (Mets-to-Death OS). In theory, if treatments for metastasis have no impact on survival (not efficacious), overall survival from initial eye treatment to death would not be changed despite the improvement of survival from metastasis to death due to lead-time bias. In this regard, it is interesting to note that the median Eye Tx-to-Death OS was also the shortest in Cohort 1 (40.8 months, 95% CI: 37.1–56.9), and similar in Cohorts 2 (62.6 months, 95% CI: 54.6–71.5) and Cohort 3 (59.4 months, 95% CI: 56.2–64.7). The Eye Tx-to-Death survival rates were consistently higher for later-treated Cohorts 2 and 3 vs. Cohort 1. Furthermore, considering the controversy on lead-time bias, we included Log Eye Tx-to-Met in the Cox proportional hazard model for Mets-to-Death OS. As shown in Table 5, this factor was not considered to be a determinant variable in Cox Models of Cohort 2 and 3. Interestingly, longer Eye Tx-to-Met was a factor for lower risk of death in Cohort 1. This might indicate that tumors discovered later might have a dormant nature. Meanwhile, the Log Eye Tx-to-Mets was not a significant predictor of Mets-to-Death OS in Cohorts 2 and 3, which may be reflective of better outcomes in patients with more aggressive tumors in the more recent time period.

Another interesting finding is the survival benefit of combined treatment modalities. In all three cohorts, patients who received liver-directed and systemic treatments (Liver-directed + Systemic) in a consecutive manner or in combination showed survival benefit over “Systemic treatment alone” or “Liver-directed treatment alone”. The interpretation of this finding is solely speculative; however, an increase in Mets-to-Death OS in proportion to an increase in the fraction of “Liver-directed + Systemic” as treatment modalities in Cohort 2 and 3 might indicate the potential role of newly emerged immunotherapies in prolonging survival of UM patients with liver metastasis. In fact, some of our liver-directed treatments have been performed with concurrent systemic immunotherapies. Itchins et al. performed a small retrospective cohort analysis of 37 patients managed in tertiary referral centers and found that transarterial chemotherapy and immunotherapy in sequence have shown to be active and reasonably well tolerated [45]. In this regard, combination treatment with immunoembolization with immune checkpoint blockades (ipilimumab and nivolumab) [NCT03472586] and radioembolization with immune checkpoint blockades [NCT02913417] are under investigation at our institution and these studies would give us more insights in the mechanism of potential survival prolongation by addition of immune checkpoint blockades to liver-directed treatments. 

Notwithstanding, our study has several limitations. First, this is a study involving retrospective analysis of survival data at a single institution. We certainly understand that the only way to scientifically prove our hypothesis is through the conduction of properly designed randomized, prospective clinical trials. However, given the high mortality of UM once it has metastasized, randomly assigning patients to liver-directed treatments or obviously inferior systemic treatments such as DTIC-based chemotherapy would not be ethically possible.

We acknowledge that since this is a retrospective analysis on “real-world” data, there might be unrecognized confounders. We also acknowledge that we are only speculating that the prolongation of survival between the cohorts is due to the treatment shift to liver-directed treatments. Earlier selection of patients suitable for liver-directed studies, methods for detection of liver metastases, better reporting of outcomes, better global management of patients over time with supportive care, multidisciplinary tumor boards, and availability of clinical trials, all have evolved and could impact overall survival of our patients. Although we did our best to adjust known confounding factors in our multivariate analysis including Eye Tx-to-Mets and Year of diagnosis and treatment of primary uveal melanoma, the above potential cofounding factors might have influenced on the survival of UM patients with hepatic metastasis. 

With the recent innovations in systemic therapies, we are now heading toward another paradigm shift on treatment of metastatic UM. Newer systemic therapies under investigation, such as Tebentafusp (IMCgp100) [NCT02570308] that have shown potential survival benefit in metastatic UM [31,33], could be combined with liver-directed therapies to further improve the outcome of UM patients with hepatic metastasis.

## 4. Materials and Methods

### 4.1. Groups Generation and Data Collection

This is a retrospective single-institution chart review on consecutive series of UM patients with liver metastasis who were treated at Thomas Jefferson University Hospital between 1971–1993 (Cohort 1, systemic therapy dominant period), 1998–2007 (Cohort 2), and 2008–2017 (Cohort 3). Cohorts 2 and 3 represent earlier and more recent decades of liver-directed therapy dominant periods. The data on patients in Cohort 1 were obtained from the database created for a previously published retrospective analysis on predictive factors for time-to-metastasis, and served as our historical control [46]. We obtained follow up survival data and updated this database. The data for Cohort 2 and 3 were obtained from a newly developed database for metastatic UM patients from 1998 with detailed survival data. All patients with liver metastasis seen and treated at Jefferson during these periods were registered into the databases. 

In Cohort 1 (1971–1993), patients were mainly treated with DTIC-based chemotherapy like cutaneous melanoma patients at the time. It was during this interval that Dr. Mavligit published his results with chemoembolization [47]. We then started exploring versions of this therapy in a limited number of patients. Over the next interval (1994–1997), we worked to standardize the liver-directed approaches. Since we do not have a complete database with survival and treatment information, and new liver-directed treatment modalities were established during this period (“transitional phase”), we excluded patients from this period. Furthermore, we arbitrarily divided the time between 1998 and 2017 into two ten-year cohorts to analyze interval changes of patient overcomes.

The following selection criteria were applied to select patients for retrospective analysis: Inclusion Criteria: Histologically confirmed liver metastasis and at least one treatment for liver metastasis (systemic, liver-directed, or both); Exclusion Criteria: Inability to obtain survival information (lost to follow-up). The following information was obtained from medical records: age, date of treatment of the primary UM, clinical features and initial therapy to the primary UM, date and site of systemic recurrence, therapy to the metastasis, and the date of death or last follow up. Liver-directed therapy included chemoembolization, immunoembolization, hepatic perfusion, plain embolization, radioembolization, drug-eluting beads, surgery, radiation, and ablative therapy. Vital status (death or survival) was confirmed by medical records as well as by reviewing public records, communication with treating physicians or by directly contacting patients or family members. The database was updated for the last time on 28 April 2018.

In terms of information on treatment modalities for liver metastasis, the first two treatment modalities after liver metastasis were collected and analyzed as “ever” treatment modalities (Treatment 1 + 2). This is based on the fact that only one patient received a third treatment modality in cohort 1 and the third or later treatment modalities were mostly palliative in nature in cohorts 2 and 3. “Ever” treatment modalities were categorized as (1) “Systemic only”, systemic treatment modalities were given as first and second treatments (if given); (2) “Liver-directed only”, liver-directed treatment modalities were given as first and second treatments (if given); (3) “Liver-directed + Systemic”, liver-directed and systemic treatments were given in consecutive order or in combination. 

Institutional Review Board (IRB)/Ethics Committee approval was obtained from Thomas Jefferson University [IRB #18D-183]. Our study adhered to the Declaration of Helsinki. Because our study was a retrospective collection of patients’ information, no informed consent was required.

### 4.2. Statistical Analysis

The primary endpoints of analysis were overall survival from the initial treatment of primary UM to death (Eye Tx-to-Death OS) and from liver metastasis to death (Mets-to-Death OS). The Kaplan–Meier curves were used to evaluate survival in each cohort. The differences in survival between cohorts was evaluated using the log-rank test adjusted for family-wise type I error of 0.05 in case of paired comparisons. Cox proportional hazards models were used to evaluate the effects of clinical-pathological factors and treatment modalities on OS endpoint in each cohort. The validity of the proportional hazard assumptions was assessed, and covariates that violated the proportional hazard assumption were included in the Cox models either as a strata variables or with the time-varying coefficient using the multiplicative hazard regression model (an extension of the Cox model that allows incorporation of non-parametric time-varying coefficients for the predictors that violate the proportional hazards assumption) [48]. The following clinical-pathological factors were considered: gender, age at initial treatment of primary uveal melanoma (<60, vs. ≥60), location of tumor (ciliary body involvement), T-stage (1–4), eye treatment (enucleation, radioactive plaque, others), adjuvant treatment indicator, and year of primary uveal melanoma diagnosis. The “ever” treatment modalities for metastases (systemic, liver-directed or both) were considered only as a predictor of Mets-to-Death OS. The Cohorts were compared in terms of these covariates using the Fisher’s Exact test generalized for more than 2 categories. The difference in continuous variable between three cohorts was analyzed using the Kruskal–Wallis test (one-way ANOVA on ranks). In addition, log-transformed time from initial eye treatment to metastasis (Log Eye Tx-to-Mets) was considered as a predictor in the Cox model for time from liver met to death (Mets-to-Death OS). Statistical analysis was performed in R (The R Foundation for Statistical Computing) [49] and SAS 9.4 (SAS Institute Inc., Cary, NC, USA).

## 5. Conclusions

To conclude, in this retrospective single center investigation, we demonstrated an improved survival between an earlier cohort and more recent cohorts of UM patients with liver metastasis. We speculate that a major contributing factor for improvement of survival is the shift of treatment modality from DTIC-based systemic chemotherapies to newly developed liver-directed treatments. We also speculate that the addition of immuno-modulatory therapies to liver-directed treatments might have improved the survival of UM patients with liver metastasis. Obviously, further investigation by way of prospective clinical trials is needed to answer these questions.

## Figures and Tables

**Figure 1 cancers-12-00117-f001:**
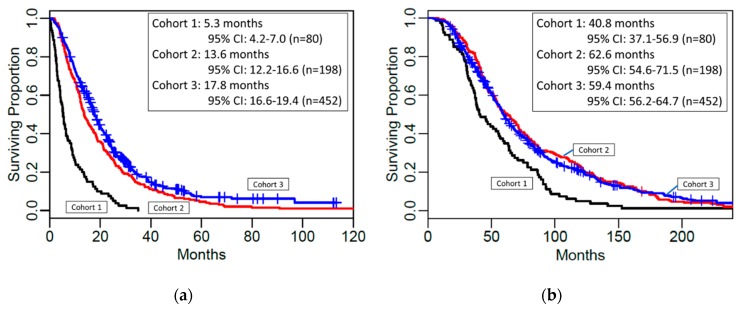
Overall survival between the three cohorts. Kaplan–Meier Curves showing (**a**) median overall survival from liver metastasis to death and (**b**) median overall survival from initial eye tumor treatment to death.

**Table 1 cancers-12-00117-t001:** Demographic Information.

**Characteristic**		**Cohort 1** **(1971–1993)**		**Cohort 2** **(1998–2007)**		**Cohort 3** **(2008–2017)**		***p*-Value**
		***N***	**%**	***N***	**%**	***N***	**%**	
**Gender**	Male	41	51%	94	47%	250	55%	0.176
	Female	39	49%	104	53%	202	45%	
AJCC Classification by T category	T 1	12	15%	19	10%	57	13%	0.283
	T 2	22	28%	36	18%	108	24%	
	T 3	25	31%	51	26%	151	33%	
	T 4	7	9%	12	6%	73	16%	
	Unknown	14	18%	80	40%	63	14%	
Tumor location	Choroid	56	70%	89	45%	286	63%	0.764
	Ciliary	23	29%	33	17%	110	24%	
	Iris	1	1%	0	0%	2	0%	
	Unknown	0	0%	76	38%	54	12%	
First Treatment for Eye Tumor	Enucleation	46	58%	52	26%	104	23%	<0.001
	Radioactive plaque	25	31%	114	58%	302	67%	
	Other	9	11%	31	16%	43	10%	
	Unknown	0	0%	1	1%	3	1%	
Adjuvant Treatment	No	75	94%	193	97%	385	85%	<0.001
	Yes (not sutent)	5	6%	5	3%	17	4%	
	Yes (sutent)	0	0%	0	0%	50	11%	
Treatment for Liver Metastasis (Treatment 1+2)	Systemic alone	56	70%	4	2%	9	2%	<0.001
	Liver-directed alone	16	20%	139	70%	251	56%	
	Liver-directed + Systemic	8	10%	55	28%	192	42%	
**Characteristic**		**Min.**	**1st Qu.**	**Median**	**Mean**	**3rd Qu**	**Max.**	***p*-Value**
Age at Eye Diagnosis	Cohort 1	22	51	60	57	66	84	0.005
	Cohort 2	19	44	53	53	62	85	
	Cohort 3	18	46	57	55	65	88	
Months from Diagnosis to Metastasis	Cohort 1	1.0	24.6	35.9	46.9	61.3	211.2	0.005
	Cohort 2	2.0	23.3	41.7	62.8	87.3	309.2	
	Cohort 3	0.3	15.7	35.6	48.9	65.6	330.7	

Min. = Minimum; Qu. = Quartile; Max. = Maximum; AJCC = American Joint Committee on Cancer 7th edition.

**Table 2 cancers-12-00117-t002:** Treatment breakdown by Treatment 1 and 2.

Treatment Group		Treatment 1	Treatment 2
Cohort 1 (*N* = 80)	Systemic alone	56	11
	Liver-directed alone	24	0
	L+S Concurrent	0	0
Cohort 2 (*N* = 198)	Systemic alone	19	39
	Liver-directed alone	178	105
	L+S Concurrent	1	2
Cohort 3 (*N* = 452)	Systemic alone	54	81
	Liver-directed alone	333	248
	L+S Concurrent	65	23

*N* = Number of patients. L+S = Concurrent liver-directed therapy and systemic therapy.

**Table 3 cancers-12-00117-t003:** Overall survival rates by cohort.

**Overall Survival Rates from Liver Metastasis**									
**Post-Mets**	**Cohort 1**			**Cohort 2**			**Cohort 3**		
**Year**	**Alive**	**LL95% CI**	**UL95% CI**	**Alive**	**LL95% CI**	**UL95% CI**	**Alive**	**LL95% CI**	**UL95% CI**
1	0.23	0.15	0.34	0.59	0.52	0.66	0.67	0.63	0.72
2	0.08	0.03	0.16	0.28	0.22	0.35	0.35	0.31	0.40
3	0	-	-	0.14	0.10	0.19	0.18	0.14	0.22
4	0	-	-	0.08	0.05	0.13	0.11	0.08	0.15
5	0	-	-	0.05	0.02	0.09	0.07	0.04	0.11
**Overall Survival Rates from Eye Treatment**									
**Post-Tx**	**Cohort 1**			**Cohort 2**			**Cohort 3**		
**Year**	**Alive**	**LL95% CI**	**UL95% CI**	**Alive**	**LL95% CI**	**UL95% CI**	**Alive**	**LL95% CI**	**UL95% CI**
2	0.84	0.76	0.92	0.91	0.88	0.95	0.88	0.85	0.91
4	0.45	0.35	0.57	0.62	0.55	0.69	0.63	0.58	0.68
6	0.26	0.18	0.38	0.42	0.36	0.49	0.40	0.35	0.45
8	0.11	0.06	0.21	0.31	0.25	0.38	0.27	0.23	0.32
10	0.05	0.02	0.13	0.23	0.18	0.30	0.21	0.17	0.25

Mets = metastasis; Tx = Eye treatment; LL = Lower limit; CI = Confidence Interval; UL = Upper limit.

**Table 4 cancers-12-00117-t004:** Median overall survival in months by cohort.

**Overall Survival from Liver Mets to Death**					
**Cohort**	**No. patients**	**No. events**	**Median OS**	**LL95% CI**	**UL95% CI**
1	80	80	5.3	4.2	7.0
2	198	197	13.6	12.2	16.6
3	452	374	17.8	16.6	19.4
**Overall Survival from Eye Tx to Death**					
**Cohort**	**No. patients**	**No. events**	**Median OS**	**LL95% CI**	**UL95% CI**
1	80	80	40.8	37.1	56.9
2	198	197	62.6	54.6	71.5
3	452	374	59.4	56.2	64.7

OS = Overall Survival; LL = Lower limit; CI = Confidence Interval; UL = Upper limit.

**Table 5 cancers-12-00117-t005:** Results from the Cox models for overall survival from liver metastasis.

**Cohort 1 (*N* = 80, 80 Events)**				
**Comparison**	**Hazard Ratio (#)**	**LL95% CI**	**UL95% CI**	***p*-value**
Eye Treatment to Metastasis Time (*)	0.83	0.70	0.97	0.022
Liver-directed only vs. Systemic	0.81	0.45	1.46	0.483
Liver-directed + Systemic vs. Liver-directed only	0.35	0.14	0.86	0.023
Liver-directed + Systemic vs. Systemic	0.28	0.12	0.65	0.003
**Cohort 2 (*N* = 194, Number of Events = 193, 4 Patients with Systemic only Tx Excluded)**				
**Comparison**	**Hazard Ratio (#)**	**LL95% CI**	**UL95% CI**	***p*-value**
Female vs. Male	0.65	0.48	0.87	0.003
Liver-directed + Systemic vs. Liver-directed only	0.58	0.42	0.80	0.001
**Cohort 3 (*N* = 443, Number of Events = 365, 9 Patients with Systemic only Tx Excluded)**				
**Comparison**	**Hazard Ratio (#)**	**LL95% CI**	**UL95% CI**	***p*-value**
Female vs. Male	0.75	0.61	0.93	0.003
Age 60+ vs. < 60	1.32	1.07	1.64	0.003
Liver-directed + Systemic vs. Liver-directed only				<0.001 (&)

(**#**) factor increase or decrease in the hazard of dying corresponding to the groups compared; (*) the reported Hazard Ratio corresponds to doubling the Eye Treatment to Metastasis Time; (&) Supremum-test of significance of time-varying coefficient in multiplicative hazard regression model. LL = Lower limit; CI = Confidence Interval; UL = Upper Limit; Tx = Treatment.

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
