# Peer review of "An Outcome Assessment of a Single Institution’s Longitudinal Experience with Uveal Melanoma Patients with Liver Metastasis"

_cancers, 2020, doi:10.3390/cancers12010117_

Round 1
Reviewer 1 Report
The manuscript entitled "An outcome assessment of a single institution’s longitudinal experience with uveal melanoma patients with liver metastasis" addresses survival after diagnosis of metastatic uveal melanoma over time indicating that some improval can be observed despite the generally low efficacy of innovative treatments inclduning targeted and immunotherapy. The paper is an important addition to this discussion. The analysis has carefully been performed and respects all statistical rules. Limits and potential biases including those inherent to a single institution's study are adequately addressed and discussed.
The discussion (in the introduction or in the discussion section) of targeted therapies could slightly be enlarged also by citing recent reviews on this topic.
Author Response
The manuscript entitled "An outcome assessment of a single institution’s longitudinal experience with uveal melanoma patients with liver metastasis" addresses survival after diagnosis of metastatic uveal melanoma over time indicating that some improval can be observed despite the generally low efficacy of innovative treatments inclduning targeted and immunotherapy. The paper is an important addition to this discussion. The analysis has carefully been performed and respects all statistical rules. Limits and potential biases including those inherent to a single institution's study are adequately addressed and discussed.
The discussion (in the introduction or in the discussion section) of targeted therapies could slightly be enlarged also by citing recent reviews on this topic.
Thank you for the comments from this reviewer. We expanded our introduction to include a more thorough review of immunotherapy and a recent review by Schank et al. published in Cancers (line 73-91). We also included a discussion of targeted therapy and a recent review by Croce et al. published in Cancers (line 92-98). Finally, the results of Tebentafusp (IMCgp100) trials and the review by Damato et al. published in Cancers have been added (line 99-108).
Reviewer 2 Report
In this manuscript, the authors performed a retrospective single-institution chart review on UM patients with liver metastasis and divided patients into 3 cohorts (cohort 1 from 1971-1993, cohort 2 from 1998-2007, cohort 3 from 2008-2017). They found that 70% of patients in cohort 1 (n=80) received only systemic therapies as their treatment option for liver metastasis, while 98% of patients in cohort 2 (n=198) and cohort 3 (n=452) received liver-directed treatment either alone or with systemic therapy. They demonstrated that both Median Mets to Death OS and Eye Tx to Death OS were shortest in cohort 1 compared to cohort 2 and 3. They then speculated that this might be due to the modern shift of treatment modalities to liver-directed therapies and combination of liver-directed and systemic treatments might further improve the survival of these patients. This paper was clearly written and did offer some insights about liver-directed therapy vs systemic treatments. It is understandable that no clinical trials could be performed to compare the clinical efficacy between liver-directed therapy vs systemic treatments. However, I have the following comments:
The authors concluded that liver-directed therapy might improve the survival of metastatic UM patients compared to systemic treatment based on the data from cohort 1-3. The systemic treatment includes both DTIC-based chemotherapy and immunotherapy. In this manuscript, systemic treatment in cohort 1 is mainly DTIC-based chemotherapy. No data are available to compare liver-directed therapy with immunotherapy alone. Therefore, the conclusion should be that liver-directed therapy could improve patient’s survival compared to DTIC-based chemotherapy instead of systemic treatments. Besides liver metastasis, do the patients in cohort 1-3 have other metastatic sites such as lung and brain? The clinical outcomes may be different if the patients have multiple metastatic sites versus liver alone. Table 5 is difficult to understand. The authors should give explanation about Hazard ratio. Does liver-directed therapy include surgical resection? In Line 297, “Groiups” should be “Group”.Author Response
In this manuscript, the authors performed a retrospective single-institution chart review on UM patients with liver metastasis and divided patients into 3 cohorts (cohort 1 from 1971-1993, cohort 2 from 1998-2007, cohort 3 from 2008-2017). They found that 70% of patients in cohort 1 (n=80) received only systemic therapies as their treatment option for liver metastasis, while 98% of patients in cohort 2 (n=198) and cohort 3 (n=452) received liver-directed treatment either alone or with systemic therapy. They demonstrated that both Median Mets to Death OS and Eye Tx to Death OS were shortest in cohort 1 compared to cohort 2 and 3. They then speculated that this might be due to the modern shift of treatment modalities to liver-directed therapies and combination of liver-directed and systemic treatments might further improve the survival of these patients. This paper was clearly written and did offer some insights about liver-directed therapy vs systemic treatments. It is understandable that no clinical trials could be performed to compare the clinical efficacy between liver-directed therapy vs systemic treatments. However, I have the following comments:
The authors concluded that liver-directed therapy might improve the survival of metastatic UM patients compared to systemic treatment based on the data from cohort 1-3. The systemic treatment includes both DTIC-based chemotherapy and immunotherapy. In this manuscript, systemic treatment in cohort 1 is mainly DTIC-based chemotherapy. No data are available to compare liver-directed therapy with immunotherapy alone. Therefore, the conclusion should be that liver-directed therapy could improve patient’s survival compared to DTIC-based chemotherapy instead of systemic treatments.
Thank you for the comments from this reviewer. We agree with this opinion and we have revised the abstract to read " DTIC-based chemotherapy" (line 29). The statement in the discussion (line 297, 301) was also changed. Our conclusion on line 445 already compared liver-directed therapy to DTIC-based systemic therapy.
Besides liver metastasis, do the patients in cohort 1-3 have other metastatic sites such as lung and brain? The clinical outcomes may be different if the patients have multiple metastatic sites versus liver alone.
We included the percent of patients with concurrent organ metastasis in the different cohorts (line 146-149). When metastasis to the liver developed, concurrent metastasis to other organs were seen in 31%, 28%, and 23% of patients in Cohort 1, 2, and 3, respectively and were comparable among the three groups. Most commonly concurrent metastasis occurred in the lung, lymph nodes, and bone. We believe that the majority of UM patients died of progression of liver metastasis, which would support the concept of liver-directed treatments for MUM patients and explain the prolonged survivals in Cohort 2 and 3.
Table 5 is difficult to understand. The authors should give explanation about Hazard ratio.
Thank you for this important critique. We agree that the concept of the hazard ratio is difficult to understand, although widely used in the medical literature. The hazard ratio for comparison of survival outcome between two groups is the ratio of the hazard function for one group and the hazard function in another group. The hazard function is just a derivative of log-transformed survival function (=probability of survival by time t) with respect to t. The hazard functions for both groups are generally varying with time, but the fundamental assumption of the Cox model is that the ratio of these functions remains constant and provides a single measure that captures the survival difference between two groups. Respectively, the standard interpretation of the hazard ratio is the factor increase or decrease in the hazard of dying. We have included such additional interpretation of some significant hazard ratios in Results (lines 224-227, 232-235). We have also added the definition of the hazard ratio in Table 5.
Does liver-directed therapy include surgical resection?
Yes, liver-directed therapy does include surgical resection. To help our readers better understand what we included as liver-directed therapy, we included a sentence in our methods section (line 399-400). Liver-directed therapy included chemoembolization, immunoembolization, hepatic perfusion, plain embolization, radioembolization, drug-eluting beads, surgery, radiation, and ablative therapy.
In Line 297, “Groiups” should be “Group”.
We have corrected this error.
Reviewer 3 Report
This manuscript retrospectively analyzes the outcomes of patients with liver metastases from uveal melanoma in 3 different cohorts from a single institution (cohort 1, from 1971-93, 80 patients; cohort 2, from 1998-2007, 198 patients and cohort 3, from 2008-2017, 452 patients). The treatment differed between the cohorts with only systemic treatment or liver directed treatment in cohort 1, and with a majority of liver directed treatment and some combined treatment in cohorts 2 and 3. The baseline characteristics of the 3 cohorts do not differ significantly, expect for the age at diagnosis, time to metastasis detection, treatment of primary tumor, adjuvant therapy and treatment of metastatic disease. The authors identified an OS from metastasis detection of 5.3 months in cohort 1, 13.6 months in cohort 2 and 17.8 months in cohort 3 and respectively an OS from primary tumor treatment to death of 40.8 months in cohort 1, 62.6 months in cohort 2 and 59.4 months in cohort 3.
This retrospective study is well conducted.
To be more complete, the recent study from Rantala et al. in melanoma research 2019 assessing the OS of 2494metastatic uveal melanoma should be mentioned at least in the introduction.
The authors rightly acknowledge that possible factors such as better and earlier detection of metastasis could have contributed to a better survival in the recent cohorts.
The authors divide treatments between systemic and liver directed. As it is shown in supplementary table 1, several different treatments have been applied to the liver. Although some data have already been published by the authors in the past, it would be interesting to know the specific outcomes of these specific treatments. Although partially explained in the introduction, the legend of supplementary table could be more detailed: which targeted therapy was tested, which checkpoints inhibitors were used?
Author Response
This manuscript retrospectively analyzes the outcomes of patients with liver metastases from uveal melanoma in 3 different cohorts from a single institution (cohort 1, from 1971-93, 80 patients; cohort 2, from 1998-2007, 198 patients and cohort 3, from 2008-2017, 452 patients). The treatment differed between the cohorts with only systemic treatment or liver directed treatment in cohort 1, and with a majority of liver directed treatment and some combined treatment in cohorts 2 and 3. The baseline characteristics of the 3 cohorts do not differ significantly, expect for the age at diagnosis, time to metastasis detection, treatment of primary tumor, adjuvant therapy and treatment of metastatic disease. The authors identified an OS from metastasis detection of 5.3 months in cohort 1, 13.6 months in cohort 2 and 17.8 months in cohort 3 and respectively an OS from primary tumor treatment to death of 40.8 months in cohort 1, 62.6 months in cohort 2 and 59.4 months in cohort 3.
This retrospective study is well conducted.
To be more complete, the recent study from Rantala et al. in melanoma research 2019 assessing the OS of 2494 metastatic uveal melanoma should be mentioned at least in the introduction.
Thank you for the comments from this reviewer. We agree with the reviewer that the review by Rantala et al. is an important pooled meta-analysis. We have added the results of this analysis to our discussion (line 276-285).
The authors rightly acknowledge that possible factors such as better and earlier detection of metastasis could have contributed to a better survival in the recent cohorts.
We appreciate these comments. To partly address this issue, we incorporated "Eye Treatment to Metastasis" into the multivariate Cox models. Interestingly, longer time from initial eye treatment to development of liver metastasis corresponded to a lower risk of dying from hepatic metastasis in Cohort 1, but not in Cohort 2 or 3. This is contradictory to the assumption that early discovery (shorter time from initial treatment to development of metastasis) increases the lead time bias and prolongs the survival of UM patients after discovery of metastasis. These results indicate that investigating the concept of "lead time bias" in a retrospective study is not simplistic. With improved screening modalities, we might discover rapidly growing liver metastasis earlier than previously; however, these tumors might be more resistant to treatments and patients might die earlier than those with longer time from primary eye treatment to systemic recurrence. This question should be addressed in carefully-designed prospective randomized clinical trials with different follow-up modalities.
The authors divide treatments between systemic and liver directed. As it is shown in supplementary table 1, several different treatments have been applied to the liver. Although some data have already been published by the authors in the past, it would be interesting to know the specific outcomes of these specific treatments.
Thank you for these important questions. We agree that the specific outcomes of different liver-directed treatments would be interesting and important to our patients. However, because the majority of our patients received different types of liver-directed and/or systemic therapy throughout their lives, it would be difficult to identify the true effect of each therapy. Even amongst the liver-directed therapies, at our institution, patients who progress may transition from one liver-directed therapy to another, based on discussions at multidisciplinary case conferences. Therefore, we decided to use general categories (liver-directed vs. systemic), rather than specific type of treatments and to not include individual treatment analysis in this manuscript.
Although partially explained in the introduction, the legend of supplementary table could be more detailed: which targeted therapy was tested, which checkpoints inhibitors were used?
We agree that it would be helpful for the reader to know what type of targeted therapy and checkpoint inhibitors have been used. We have updated the legend of supplementary Table 1 to identify what therapies were used. Since the number of patients in individual targeted therapies is small, we did not separately analyze the effects of individual treatment in this retrospective study.

Round 2
Reviewer 2 Report
The authors addressed all my concerns.